# Dietary Amino Acid Composition and Glycemic Biomarkers in Japanese Adolescents

**DOI:** 10.3390/nu16060882

**Published:** 2024-03-19

**Authors:** Masayuki Okuda, Satoshi Sasaki

**Affiliations:** 1Graduate School of Sciences and Technology for Innovation, Yamaguchi University, 1-1-1 Minami-Kogushi, Ube 755-8505, Japan; 2Department of Social and Preventive Epidemiology, School of Public Health, The University of Tokyo, 7-3-1 Hongo, Tokyo 113-0033, Japan

**Keywords:** adolescents, amino acids, compositional data analysis, dietary assessment, glucose, HOMA index, isoleucine, insulin, leucine, methionine

## Abstract

Protein intake reportedly increases the risk of diabetes; however, the results have been inconsistent. Diabetes in adulthood may be attributed to early life dietary amino acid composition. This study aimed to investigate the association between amino acid composition and glycemic biomarkers in adolescents. Dietary intake was assessed using a food frequency questionnaire, and fasting glucose and insulin levels were measured in 1238 eighth graders. The homeostatic model assessment (HOMA) indices (insulin resistance and β-cell function) were calculated. Anthropometrics were measured and other covariates were obtained from a questionnaire. Amino acid composition was isometric log transformed according to the compositional data analysis, which was used as explanatory variables in multivariate linear regression models for glucose, insulin, and HOMA indices. Only the association between glucose and leucine was significant. In replacement of other amino acids with leucine, an increase of 0.1% of total amino acids correlated with a lower glucose level (−1.02 mg/dL). One-to-one substitution of leucine for isoleucine or methionine decreased glucose (−2.98 and −2.28 mg/dL, respectively). Associations with other biomarkers were not observed. In conclusion, compositional data analysis of amino acids revealed an association only with glucose in adolescents; however, the results of this study should be verified in other populations.

## 1. Introduction

Diabetes is a morbidity and economic burden on society. Diabetes and prediabetes, which are a continuum of impaired insulin resistance and/or secretion, develop even in adolescents [1]. Impaired glucose metabolism in adolescents is associated with increased cardiovascular risk [2]. The influence of dietary behaviors on type 2 diabetes (T2D) incidence has been documented extensively in the literature [3]. One meta-analysis showed no association between diabetes and total dietary fat and carbohydrates as macronutrients [3]; however, the intake of protein, especially animal protein, has been reported to be positively associated with diabetes incidence in linear dose–response relationships [4,5]. Nonetheless, evidence on the effects of protein intake does not always support this hypothesis [6]. Plant proteins, a counterpart of animal proteins, show no or U-shaped associations with diabetes [4,5]. Furthermore, randomized controlled trials on changes in total protein intake or the replacement of animal with plant proteins have not offered any positive evidence [6]. Inconsistent results have been observed for several food groups that provide animal proteins, such as fish, dairy products, and eggs [5,7,8,9]. These results suggest that different amino acid compositions may play a role in glycemic homeostasis.

Single or grouped dietary amino acids have been investigated for their roles in T2D and glucose metabolism. Branched-chain amino acids (BCAAs), such as leucine, isoleucine, and valine, are potential contributors to T2D [10,11,12] and gestational diabetes [13]; however, the results from an adult Japanese cohort contradicted this [14]. Amino acids other than BCAAs are also involved in glucose metabolism. Excess intake of sulfur amino acids (methionine and cysteine) is associated with a higher risk of diabetes in the US population [15], and histidine is negatively associated with glucose and insulin resistance in the Chinese population [16]. In diabetic rats, alanine and glutamine supplementation reduces and improves blood glucose levels [17]. However, to the best of our knowledge, the effects of amino acid intake on glucose metabolism in adolescents have rarely been reported. Moreover, the effects of amino acid combinations are unknown.

The intake of amino acids is derived from proteins composed of 20 amino acids. The amount of each amino acid in a protein influences the proportion of other amino acids. For example, if a protein has a high amount of one type of amino acid, it may have a low amount of another. In this context, single amino acids as independent variables in linear regression models cannot sufficiently explain the association with health risks because other amino acids concurrently change in composition. One solution is to use compositional data analysis, whereby parameters are expressed as a log ratio between components and are used as variables of the entire composition. Three compositional data analyses have been applied to nutritional epidemiology: energy intake [18], chrono-nutrition [19], and fat intake [20].

Most previous studies have targeted adult populations, and the incidence of T2D was evaluated in a prospective cohort study. Insulin resistance and prediabetes incidence were associated with protein intake in adults [8]. Therefore, a predisposition to insulin resistance may emerge in early life. This could be observed in a cross-sectional study of adolescents because dietary behaviors can be tracked from childhood and adolescence to adulthood [21,22]. Hence, this study aimed to investigate the association between dietary amino acid composition and glycemic biomarkers in adolescents.

## 2. Materials and Methods

### 2.1. Participants

Eighth-grade students attending 1 of all 16 junior high schools in Shuna City, Japan participated in the Shunan Healthy Diet for Children Project and the Shunan Child Cohort study, from 2006 to 2008, in a method described previously [23,24]. The students were encouraged to participate in the project organized by the Shunan City educational board and schools, and both they and their guardians provided written informed assent and consent, respectively, to participate in this study, allowing for the collection of their samples of their own volition. Blood samples were drawn after 10 h of fasting to measure plasma glucose, serum lipid, and liver enzyme levels. The serum remaining after the laboratory tests was stored at −80 °C until further analysis for insulin by the chemiluminescent enzyme immune assay of SRL, Inc. (Tokyo, Japan). Height and weight were measured at school during annual medical checkups. The participants completed questionnaires regarding their dietary and lifestyle habits.

### 2.2. Amino Acids

Dietary foods and nutrients were assessed using the brief self-administered diet history questionnaire for youths (BDHQ15y), which was revised and modified from the Diet History Questionnaire (DHQ) [25] and BDHQ for adults [26]. The habitual intake of 80 foods and condiments during the previous month was calculated through 90 items. Nutrients were estimated based on the Standard Tables of Food Composition in Japan 2010 [27] and the amino acid database developed by Suga et al. [28], which differentiates amino acids between 18 kinds: isoleucine, leucine, lysine, methionine, cysteine, phenylalanine, tyrosine, threonine, tryptophan, valine, histidine, arginine, alanine, aspartic acid, glutamic acid, glycine, proline, and serine. Asparagine and glutamine are involved in the synthesis of aspartic acid and glutamic acid, respectively. Using the DHQ, Pearson’s correlations of energy-adjusted amino acids with those calculated from dietary records ranged from 0.21 to 0.50 for men and 0.35 to 0.58 for women [29]. Using the BDHQ15y for adolescents, protein intake was estimated with relative validity in Pearson’s correlations from 0.109 to 0.302 for urinary nitrogen excretion [30]. Plausible answers to the BDHQ were evaluated as energy intake between half of the age- and sex-specific estimated energy requirements for low physical activity levels [31] and 1.5 times those for high physical activity levels. Amino acid composition was expressed as a percentage of the total amino acids. The intake of other nutrients was expressed as nutrient density (% of total energy intake, %E, or the standardized amount of estimated energy requirement).

### 2.3. Glucose Metabolism

In addition to glucose (mg/dL) and insulin (μU/mL), the homeostatic model assessment for insulin resistance (HOMA-IR) and β-cell function (HOMA-β) were calculated as insulin (μU/mL) × glucose (mmol/L)/22.5 and 20 × insulin (μU/mL)/(glucose [mmol/L] − 3.5), respectively [32], in which glucose (mg/dL) was multiplied by 0.05551 for unit conversion to mmol/L [33].

### 2.4. Covariates

Body mass index (BMI, kg/m^2^) was calculated as (weight [kg]/height [cm]^2^) × 10,000. BMI was sex- and age-standardized as a standard deviation score (zBMI) using the Lamda–Mu–Sigma method [34] with the Japanese 2000 reference [35]. Age in months was calculated as the difference between the birth and measurement date. Data on physical activity (>2 times/week), sleep duration (h/d), screen time (h/d), single parents, passive smoking, and siblings (one, two, three, or more) were obtained from the lifestyle questionnaire.

### 2.5. Statistical Analysis

The participant selection process is shown in Figure 1. Of the 4101 students who attended junior high schools, 3524 students gave informed consent. Serum samples from 44.5% (1447/3254) of adolescents who participated in this study were available for insulin measurement. After excluding participants who did not answer either questionnaire (*n* = 15), had implausible energy intake (*n* = 40), missed items in the questionnaire (*n* = 150), or had physician-diagnosed diseases (any of diabetes mellitus, dyslipidemia, hypertension, heart disease, and kidney disease; *n* = 4), data from 1238 healthy participants were analyzed. Amino acids are represented as geometric means and covariances between each pair, arithmetic means, and standard deviations (SDs). The distribution of outcome variables, e.g., fasting plasma glucose, serum insulin, HOMA-IR, and HOMA-ß, were checked using quantile–quantile plots if they had normality. Insulin, HOMA-IR, and HOMA-ß had skewed distributions; hence, they were natural log transformed and presented as geometric means with 95% confidence intervals (CIs). Other variables are presented as arithmetic means and SDs for continuous variables, counts, and percentages for categorical variables.

Associations between the amino acid composition and outcome variables were examined using multivariate linear regression models. Amino acid composition was isometric log transformed (ILR) for independent variables in the regression models. The coefficient of the first amino acid in the order of amino acids represented the effect of the ratio of the first amino acid to other amino acids; therefore, the regression analysis was repeated 18 times by changing the orders (Appendix A). Changes in predictive values were calculated when 0.1% of the total amino acids were replaced from one amino acid to another. The geometric mean of the amino acids was used as the reference for replacement. Two methods of replacement were used: one-to-all other amino acid replacement, that is, a 0.1% increase in one amino acid and one-seventeenth of a 0.1% decrease in the other 17 amino acids; and one-to-one replacement, that is, a 0.1% substitution between each pair of amino acids. A 0.1% replacement was selected as a realistic value based on the SD of the proportion of amino acids. The CIs of the substitution effect were calculated using two standard errors of the predicted values. Multivariate regression models were used to control for covariates. Nutrients such as energy, protein, saturated fatty acids, total dietary fiber, and glycemic load may be possible confounders. The inclusion of these factors in the multivariate models was determined using a simple linear regression analysis for glucose and log-transformed insulin. Age, sex, zBMI, and other lifestyle variables were included in the multivariate models. The linear model assumption was verified visually for linearity, normality, and homoscedasticity. Variation inflation factors were calculated to check for multicollinearity. Additional linear regression analysis was conducted for grouped amino acids. Amino acids were classified based on amino acid side chains: BCAAs, aromatic (phenylalanine, tyrosine, and tryptophan), sulfur, and other amino acids (lysine, threonine, histidine, arginine, alanine, aspartic acid, glutamic acid, glycine, proline, and serine). R version 4.3.1 and compositions, tidyverse, and DescTools packages were used for statistical analyses [36,37,38], and the significance level was set at 0.05.

## 3. Results

### 3.1. Participants

The participants included 651 males (52.6%) and 587 females (47.4%), with a mean age of 163.5 ± 3.4 months (13.6 ± 0.3 years). The mean BMI was 19.2 ± 2.6 kg/m^2^ (zBMI: −0.25 ± 0.90), and the number of participants with a zBMI of more than 1.5 was 42 (3.4%). The mean fasting plasma glucose level was 90.6 ± 5.6 mg/dL, with the maximum being 110 mg/dL. The geometric mean of fasting serum insulin was 6.19 μU/mL (95% CI: 6.03–6.35) and the maximum was 21.4 μU/mL. The other participant characteristics are shown in Table 1.

### 3.2. Amino Acid Intake

The mean amino acid intake is shown in Table 2; the geometric means are similar to the arithmetic ones with coefficients of variance of 0.018–0.091. Glutamic acid was the most abundant amino acid (18.68%), followed by aspartic acid, leucine, and lysine (9.36%, 8.24%, and 6.85%, respectively). The covariance between each pair of amino acids is shown in Appendix A.

### 3.3. Regression Analysis

The intakes of energy, protein, and total dietary fiber were significantly associated with glucose levels in simple regression models (Appendix A) and were included in the multivariate models. Among the linear multivariate regression analysis of the amino acid composition for plasma glucose, serum insulin, HOMA-IR, or HOMA-ß, only leucine as the first term of the composition was significantly associated with plasma glucose (*p* = 0.025; Table 3). When amino acid groups instead of individual amino acids were used as independent variables, no significant association was found (Appendix A).

### 3.4. Replacement of Amino Acids

When 0.1% of the total amino acids were replaced with other amino acids, an increase in leucine from the mean was significantly associated with low glucose levels by −1.02 mg/dL (Table 4). In the ILR transformation, the sign of the number was reversed, so the coefficient of leucine in Table 3 was positive, and the effect of leucine in Table 4 was negative. In contrast, increased isoleucine and methionine were associated with high glucose levels (2.09 and 1.34 mg/dL, respectively). In a one-to-one substitution of amino acids, replacement of 0.1% of total protein to leucine from isoleucine, lysine, methionine, phenylalanine, tyrosine, alanine, aspartic acid, glutamic acid, glycine, or proline resulted in low glucose levels (−2.98 mg/dL for isoleucine, −2.28 mg/dL for methionine, and −1.94 to −0.81 mg/dL for the following: Figure 2a and the second row in Appendix A), while 0.1% substitution of these amino acids for leucine showed high glucose levels (Figure 2b and the second column in Appendix A). Glucose differences regarding the increase and decrease in leucine were asymmetrical. Isoleucine and methionine showed similar effects in one-to-one and one-to-all replacements. Histidine showed a significant effect on glucose levels in the replacement with phenylalanine and tyrosine, in addition to isoleucine and methionine, but did not affect the one-to-all replacement. Serum insulin, HOMA-IR, and HOMA-ß were not significantly influenced by amino acid compositions (Table 3 and Table 4 and Appendix A). Linearity, normality, and homoscedasticity were assumed in the models, and the maximum variation inflation factor was 2.61 for protein.

## 4. Discussion

The dietary intake of amino acids in Japanese adolescents was examined using compositional data analysis, and its associations with glycemic biomarkers were explored. We found that adolescents with a high leucine intake relative to other amino acids, especially when substituting isoleucine and methionine, had lower glucose levels. In contrast, isoleucine and methionine intake were associated with high glucose levels. The composition of dietary amino acid indices did not show significant associations with insulin or HOMA indices.

Compositional data analysis was applied to data with constant sum constraints, such as the chemical composition of food and geology, time allocation of daily activity, household expenditure, income, etc. [39,40,41]. Hence, a scatter plot of any pair of components shows an aggregated distribution but not a linear association or random distribution. This feature makes the results difficult to interpret as a dose–response relationship around a threshold level. When applying compositional data analysis, the effect may be estimated to be influenced by shifting the proportion of dietary amino acids from the mean level, which can propose a direction for public health policy.

Isoleucine, leucine, and valine are BCAAs that have been extensively investigated in animal and human glucose metabolism. In an obese mouse model, a BCAA-rich diet had favorable effects on serum glucose levels and glucose tolerance [42]. Mixed BCAA supplementation also improved glucose intolerance in mouse models [43]. A single oral dose and chronic supplementation of leucine increased insulin sensitivity [44,45], and the oral administration of isoleucine, but not leucine, partially inhibited plasma glucose elevation before the glucose load [46]. Insulin levels and HOMA indices were not influenced by dietary amino acids in this study; however, this may be because the supplementary administration in animals did not exactly imitate the complexity of the ingredients in a usual human diet. Nonetheless, the effects of BCAAs have been inconsistent in epidemiological studies. In US studies of adult female cohorts, the incidence of diabetes and gestational diabetes was positively associated with dietary BCAAs [10,11,13]. However, dietary BCAAs were not associated with glucose, insulin, or HOMA-IR in overweight Iranian adults [47]. Furthermore, a higher intake of BCAAs, especially leucine, was negatively associated with the incidence of diabetes in a Japanese cohort [14], but the intake of isoleucine was not. This cross-sectional study of Japanese adolescents also demonstrated a negative association between leucine and glucose levels and an interrelation between leucine and isoleucine. A previous study on Japanese adults partially supports the results of this study. The mechanistic targets of rapamycin complexes are integrators of signaling from amino acids and glucose and are most sensitive to leucine [48]. Hence, leucine and isoleucine may exert different biological effects. Based on the results of this study, it may be difficult to discuss the effects of all BCAAs.

Dietary methionine restriction in rats enhances lifespan and improves glucose metabolism [49]. Basal insulin and glucose levels were decreased by chronic methionine restriction in rats [50]. Dietary methionine restriction in healthy human adults does not significantly alter insulin or glucose levels; however, the plasma levels of BCAAs, especially leucine and valine, decrease [51]. It is difficult to interpret the discrete effects of a single amino acid on glucose metabolism. Sulfur amino acids, including methionine and cysteine, are rich in egg and meat products, low in plant proteins, and consumed more in a high-protein dietary pattern than in a vegetable dietary pattern [52]. Systematic reviews have suggested that animal protein increases the risk of diabetes and plant protein decreases it [6], and that replacing animal protein with plant protein decreases risk [53]; however, the evidence is insufficient. The results of this study are not decisive in recommending methionine restriction because of the growth retardation seen in mice [49].

Histidine supplementation significantly improved HOMA-IR and adiposity in obese women aged between 33 and 51 years with metabolic syndrome compared with baseline values and the control group in a randomized controlled trial [54]. Fasting blood glucose and serum insulin levels decreased after the intervention; however, the effects were not significant. The authors suggested that elevated serum histidine levels could independently improve insulin sensitivity. In a cross-sectional study of Chinese adults, histidine intake, estimated using an online food frequency questionnaire, showed a significantly negative association with fasting blood glucose and HOMA-IR [16]. The association with insulin was negative but not significant. The associations between histidine and glucose levels or HOMA-IR were not significant in this study.

This study had several limitations. First, the cross-sectional design limited interpretation. Although cohort studies have revealed the incidence of T2D, insulin resistance is thought to precede the development of T2D by 10 to 15 years [55]. However, it is unclear whether the high glucose levels related to amino acid composition in this study could be related to future insulin resistance in adolescents. The blood glucose levels measured in this study were within the normal range for adults [56]; however, glucose, insulin, and HOMA levels increase with age in childhood and adolescence [57,58]. Early pathophysiology could have appeared in adolescents since dietary patterns in youth track to dietary habits in adulthood [21,22]. In an experimental study in rats, fasting hyperglycemia preceded insulin resistance after a sub-chronic high-fructose diet [59]. Further studies with a long follow-up period are needed to elucidate the pathophysiological implications of amino acid compositions on blood glucose levels in Japanese adolescents and those of other ethnicities. In addition, a follow-up study in this population is expected to elucidate the effects on other glycemic biomarkers and the incidence of prediabetes and diabetes.

Second, compositional data analysis has some common limitations. The results suggested that leucine was favorably involved and isoleucine and methionine were unfavorably involved in glucose metabolism; however, this does not mean that each of these amino acids has direct biological effects. These effects were determined by the ratio of amino acids to other amino acids. As the effects were observed only around the mean composition, they are not applicable to extreme compositions.

Third, a multiple comparison problem occurs when repeating an analysis for each amino acid. There is a chance of false detection of a non-significant effect in the sample. The result from the linear regression analysis was not inconsistent with that from one-to-all and one-to-one replacement of amino acids. To reduce the number of variables, amino acids can be grouped, but the mechanism of action is not explainable separately based on any type of classification.

Finally, the dietary assessment used in this study may have had measurement errors similar to other food frequency questionnaires. Although social preference bias may influence the selection between animal and plant proteins, it may not influence the composition of amino acids, which were not detectable by the participants.

## 5. Conclusions

We found an association between the composition of dietary amino acids and fasting blood glucose levels in adolescents. In particular, a high leucine intake relative to other amino acids was associated with lower glucose levels. Additionally, isoleucine and methionine exhibited the opposite effect on glucose levels. The effects on glucose varied among amino acids within the same group of BCAAs, which could not simply explain the collective effects of amino acid groups, such as BCAAs. Therefore, the effects of each amino acid on health risks should be distinctively investigated. Given that the proportions of amino acids in the diet are influenced by each other, compositional data analysis could reveal causality between amino acids and health status by considering the interrelationship of amino acids. These findings underscore the importance of understanding amino acid composition in dietary management and preventive strategies for metabolic disorders in adolescents. In this study of Japanese adolescents, only the effect on glucose, but not the other biomarkers, was observed; thus, the causal relationship to diabetes remains unclear. Future studies in other populations with different dietary patterns and a follow-up study in this population should be conducted to elucidate the effects of amino acid composition.

## Figures and Tables

**Figure 1 nutrients-16-00882-f001:**
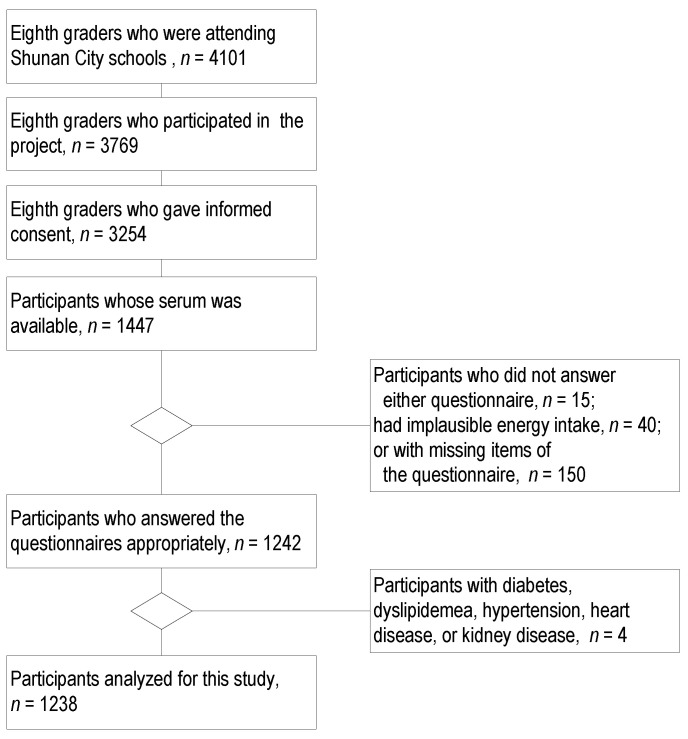
Flowchart of participant selection.

**Figure 2 nutrients-16-00882-f002:**
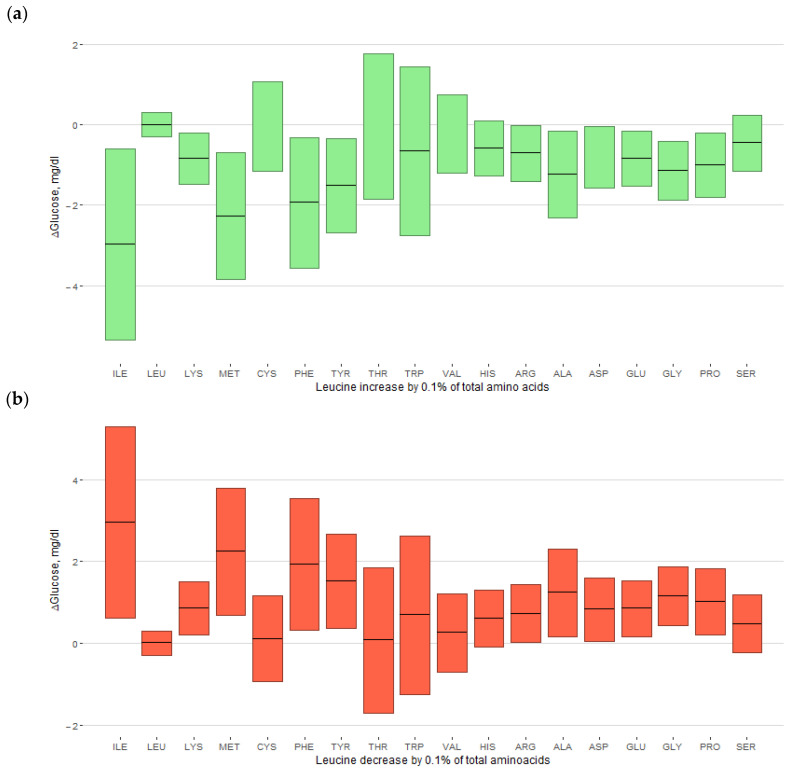
Estimated glucose difference (ΔGlucose) in one-to-one replacement by 0.1% of total amino acids. For (**a**) the replacement of leucine with each of the other amino acids and (**b**) the replacement of each amino acid with leucine, estimates (bars) and 95% confidence intervals (boxes) are plotted. ILE, isoleucine; LEU, leucine; LYS, lysine; MET, methionine; CYS, cysteine; PHE, phenylalanine; TYR, tyrosine; THR, threonine; TRP, tryptophan; VAL, valine; HIS, histidine; ARG, arginine; ALA, alanine; ASP, aspartic acid; GLU, glutamic acid; GLY, glycine; PRO, proline; SER, serine.

**Table 1 nutrients-16-00882-t001:** Participant characteristics, *n* = 1238.

	Mean, Geometric Mean *, *n*	SD, 95% CI *, (%)
Sex: Male	651	(52.6%)
Female	587	(47.4%)
Age, months	163.5	3.4
BMI, kg/m^2^	19.2	2.6
zBMI	−0.25	0.90
Exercise ≥3 times/week	947	(76.5%)
Screen time, h	4.2	1.0
Sleep duration, h	7.4	0.7
Energy, kcal	2266.4	634.9
Protein, %E	14.2	2.3
Total dietary fiber, g/1000 kcal	5.5	1.4
Saturated fatty acids, %E	10.6	2.5
Glycemic load, /1000 kcal	68.4	22.5
Single parent	71	(5.7%)
Siblings: 1	118	(9.5%)
2	588	(47.5%)
≥3	532	(43.0%)
Passive smoking in household	638	(51.5%)
Plasma glucose, mg/dL	90.6	5.6
Serum insulin, μU/mL	6.19 *	6.03, 6.35 *
HOMA-IR	1.38 *	1.34, 1.42 *
HOMA-β, %	82.8 *	80.8, 84.9 *

SD, standard deviation; BMI, body mass index; zBMI, sex- and age-standardized BMI; HOMA-IR and HOMA-β, homeostatic model assessment for insulin resistance and beta cell function, respectively. * Geometric mean and 95% confidence interval (CI).

**Table 2 nutrients-16-00882-t002:** Dietary amino acids ^1^.

	Geometric Mean, %	Arithmetic Mean ± SD, %
Isoleucine	4.56	4.56 ± 0.10
Leucine	8.24	8.24 ± 0.18
Lysine	6.85	6.85 ± 0.43
Methionine	2.47	2.47 ± 0.09
Cysteine	1.55	1.55 ± 0.10
Phenylalanine	4.66	4.66 ± 0.09
Tyrosine	3.66	3.66 ± 0.07
Threonine	4.10	4.10 ± 0.10
Tryptophan	1.23	1.23 ± 0.03
Valine	5.39	5.39 ± 0.13
Histidine	3.47	3.47 ± 0.23
Arginine	5.65	5.64 ± 0.36
Alanine	4.99	4.99 ± 0.27
Aspartic acid	9.36	9.36 ± 0.36
Glutamic acid	18.68	18.68 ± 0.90
Glycine	4.21	4.21 ± 0.30
Proline	6.10	6.10 ± 0.55
Serine	4.84	4.84 ± 0.13

^1^ Amino acids are expressed as percentages of total amino acids. SD, standard deviation.

**Table 3 nutrients-16-00882-t003:** Regression coefficients of the first term relative to the other terms in ILR compositions.

	Glucose, mg/dL		log(Insulin, μU/mL)		log(HOMA-IR)		log(HOMA-β, %)	
	β (SE)	*p*	β (SE)	*p*	β (SE)	*p*	β (SE)	*p*
Isoleucine	−119.1 (64.2)	0.066	0.78 (4.89)	0.818	−0.56 (5.18)	0.969	5.56 (4.68)	0.209
Leucine	50 (22.6)	0.025	1.01 (1.72)	0.564	1.57 (1.82)	0.393	−0.95 (1.64)	0.537
Lysine	26.5 (27.3)	0.326	−0.47 (2.07)	0.792	−0.17 (2.20)	0.913	−1.59 (1.99)	0.399
Methionine	−32.6 (18.9)	0.073	0.11 (1.44)	0.926	−0.26 (1.52)	0.868	1.45 (1.37)	0.268
Cysteine	42.8 (31.8)	0.159	−0.51 (2.42)	0.837	−0.04 (2.57)	1.000	−2.14 (2.32)	0.337
Phenylalanine	−17.9 (32.8)	0.511	0.58 (2.49)	0.862	0.38 (2.64)	0.943	1.32 (2.39)	0.574
Tyrosine	−40.5 (33.1)	0.188	0.51 (2.52)	0.846	0.04 (2.67)	0.996	2.19 (2.41)	0.338
Threonine	22.6 (36.5)	0.468	−0.72 (2.77)	0.825	−0.45 (2.94)	0.920	−1.78 (2.66)	0.488
Tryptophan	−23.3 (27.1)	0.430	1.23 (2.06)	0.518	0.97 (2.18)	0.618	2.21 (1.97)	0.259
Valine	17.7 (23.3)	0.480	−0.98 (1.77)	0.549	−0.78 (1.88)	0.643	−1.75 (1.70)	0.294
Histidine	0.2 (7.0)	0.890	0.06 (0.53)	0.833	0.06 (0.56)	0.826	0.05 (0.51)	0.891
Arginine	19 (19.2)	0.283	−0.79 (1.46)	0.615	−0.58 (1.55)	0.746	−1.55 (1.40)	0.261
Alanine	−17.6 (17.1)	0.262	0.64 (1.30)	0.686	0.44 (1.38)	0.824	1.35 (1.25)	0.289
Aspartic acid	−2 (14.0)	0.837	−0.59 (1.07)	0.569	−0.62 (1.13)	0.568	−0.45 (1.02)	0.674
Glutamic acid	15.4 (12.5)	0.249	1.25 (0.95)	0.212	1.42 (1.01)	0.182	0.67 (0.91)	0.474
Glycine	−3.1 (13.7)	0.924	−0.06 (1.04)	0.946	−0.09 (1.10)	0.955	−0.03 (0.99)	0.982
Proline	−19.2 (19.0)	0.305	−0.22 (1.45)	0.852	−0.44 (1.53)	0.749	0.57 (1.38)	0.693
Serine	81.3 (49.9)	0.109	−1.81 (3.80)	0.586	−0.89 (4.02)	0.773	−5.13 (3.63)	0.142

Insulin, homeostatic model assessment for insulin resistance (HOMA-IR) and for beta cell function (HOMA-β) were natural log transformed. ILR; isometric log transformed; β, coefficient; SE, standard deviation. Multivariate regression models were adjusted for age, sex, a z-score of body mass index, physical activity, sleep duration, screen time, single parents, passive smoking, siblings, energy, protein, and total dietary fiber.

**Table 4 nutrients-16-00882-t004:** The effect of amino acid increases by 0.1% of total amino acids in the replacement of the other 17 amino acids.

	Glucose, mg/dL		log(Insulin, μU/mL)		log(HOMA-IR)		log(HOMA-β)	
	Effect	95%CI	Effect	95%CI	Effect	95%CI	Effect	95%CI
Isoleucine	2.09	(0.11, 4.07)	−0.05	(−0.20, 0.10)	−0.02	(−0.18, 0.14)	−0.13	(−0.28, 0.01)
Leucine	−1.02	(−1.76, −0.28)	−0.01	(−0.07, 0.04)	−0.02	(−0.08, 0.04)	0.03	(−0.03, 0.08)
Lysine	−0.13	(−0.8, 0.55)	0.01	(−0.04, 0.06)	0.01	(−0.05, 0.06)	0.01	(−0.04, 0.06)
Methionine	1.34	(0.18, 2.5)	−0.01	(−0.09, 0.08)	0.01	(−0.08, 0.1)	−0.06	(−0.15, 0.02)
Cysteine	−0.92	(−2.13, 0.3)	0.00	(−0.09, 0.09)	−0.01	(−0.11, 0.09)	0.03	(−0.05, 0.12)
Phenylalanine	1.01	(−0.26, 2.29)	−0.02	(−0.11, 0.08)	−0.01	(−0.11, 0.1)	−0.05	(−0.15, 0.04)
Tyrosine	0.58	(−0.25, 1.40)	0.00	(−0.06, 0.07)	0.01	(−0.06, 0.08)	−0.02	(−0.08, 0.04)
Threonine	−0.95	(−2.7, 0.79)	0.02	(−0.11, 0.15)	0.01	(−0.13, 0.15)	0.06	(−0.06, 0.19)
Tryptophan	−0.30	(−2.43, 1.82)	−0.02	(−0.18, 0.14)	−0.02	(−0.19, 0.15)	−0.01	(−0.16, 0.15)
Valine	−0.76	(−1.76, 0.24)	0.03	(−0.05, 0.11)	0.02	(−0.06, 0.10)	0.06	(−0.01, 0.13)
Histidine	−0.39	(−0.81, 0.03)	0.00	(−0.03, 0.04)	0.00	(−0.03, 0.03)	0.02	(−0.01, 0.05)
Arginine	−0.26	(−0.73, 0.21)	0.00	(−0.03, 0.04)	0.00	(−0.04, 0.04)	0.01	(−0.02, 0.05)
Alanine	0.28	(−0.44, 1.00)	−0.02	(−0.07, 0.04)	−0.02	(−0.07, 0.04)	−0.03	(−0.08, 0.02)
Aspartic acid	−0.16	(−0.54, 0.22)	0.00	(−0.03, 0.03)	0.00	(−0.03, 0.03)	0.01	(−0.02, 0.03)
Glutamic acid	−0.63	(−1.63, 0.37)	−0.02	(−0.1, 0.05)	−0.03	(−0.11, 0.05)	0.00	(−0.07, 0.08)
Glycine	0.90	(−1.13, 2.93)	0.11	(−0.04, 0.27)	0.12	(−0.04, 0.29)	0.09	(−0.06, 0.24)
Proline	0.18	(−1.13, 1.49)	0.07	(−0.03, 0.17)	0.07	(−0.03, 0.18)	0.06	(−0.03, 0.16)
Serine	−2.54	(−5.76, 0.67)	0.06	(−0.19, 0.30)	0.03	(−0.23, 0.29)	0.16	(−0.07, 0.39)

Insulin, homeostatic model assessment for insulin resistance (HOMA-IR) and for beta cell function (HOMA-β) were natural log transformed. CI, confidence interval.

## Data Availability

The data presented in this study are available upon request from the corresponding author. These data are not publicly available because of privacy concerns.

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
