# Peer review of "Dietary Amino Acid Composition and Glycemic Biomarkers in Japanese Adolescents"

_nutrients, 2024, doi:10.3390/nu16060882_

Round 1
Reviewer 1 Report
Comments and Suggestions for Authors
Comments
The manuscript titled “Dietary Amino Acid Composition and Glycemic Biomarkers in 2 Japanese Adolescents” is well written. For some time now, we in the 21st century have been aware of the importance of reducing the intake of carbohydrates for health, especially simple or rapidly absorbed carbohydrates. For this reason, there are many specialists who recommend diets rich in protein. Thus, I consider that the manuscript is current and is focused on solving problems of today's society.
The results are well presented in Table format. Although I have some suggestions for authors:
Minors:
1.-The results section could appear structured in a similar way that in the material and methods section. By adding subsections, this can help the reader follow these results.
2.-In the supplementary material they provide, figures S1-4, they are again tables with heat diagrams. Although the tables clearly reflect what is stated in the results and what is discussed in the discussion section, it would be pertinent for them to reflect at least the results with statistical significance in dot-plot diagrams, bar diagrams, etc. so that it is more illustrative.
Furthermore, the file "AminoAcidsComp20240128" is missing, since this is again the manuscript in word format. Likewise, the file "Supplementary_TableS1" cannot be opened.
3.-Although the authors clearly explain the limitations of the study, it would be interesting that in addition to reproducing these studies in another population, it would be interesting to repeat the study after a certain time, on the same population, that is, to carry out prospective longitudinal studies.

Author Response
March 11th 2024
Dear Reviewer 1,
We appreciate your comments with positive suggestions and advice. We have responded to each of the comments, added new figures, and amended the manuscript and supplementary files, in conjunction with considering other reviewer’s comments. Corrected sentences have been yellow-highlighted in the revised manuscript.
I, on behalf of the coauthors, wish to resubmit the manuscript. We hope that the manuscript is revised according to the reviewers’ intentions and is now suitable for publication.
Sincerely,
Masayuki Okuda
1.-The results section could appear structured in a similar way that in the material and methods section. By adding subsections, this can help the reader follow these results.
Response: Thank you for your advice. Accordingly, to improve the organization of the manuscript, we have added subheadings in the Results section as follows: 3.1. Participants, 3.2. Amino acid intake, 3.3. Regression analysis, and 3.4. Replacement of amino acids.
2a.-In the supplementary material they provide, figures S1-4, they are again tables with heat diagrams. Although the tables clearly reflect what is stated in the results and what is discussed in the discussion section, it would be pertinent for them to reflect at least the results with statistical significance in dot-plot diagrams, bar diagrams, etc. so that it is more illustrative.
Response: Thank you for your suggestion. We have added Figure 3, which illustrates the effect of one-to-one replacement on glucose levels with respect to leucine, which showed a significant association in one-to-all replacement (Table 4). The figure numbers have been cited in Lines 212, and 214, and supporting information have added in Lines 214–2215, “Glucose differences on increase and decrease in leucine were asymmetrical.” Supplementary Figures S1–S4 have been retained to present numerical values.
2b.-Furthermore, the file “AminoAcidsComp20240128” is missing, since this is again the manuscript in word format. Likewise, the file “Supplementary_TableS1” cannot be opened.
Response:We apologize for the inconvenience caused by these technical issues. The file, “AminoAcidsComp20240128,” has been deleted from the zipped folder as it was duplicated. Additionally, the file “Supplementary _TableS1,” which is the main R script, has been converted to a .docx file. We hope it can now be accessed.
3.-Although the authors clearly explain the limitations of the study, it would be interesting that in addition to reproducing these studies in another population, it would be interesting to repeat the study after a certain time, on the same population, that is, to carry out prospective longitudinal studies.
Response: Thank you for your valuable suggestion. We acknowledge that our results could not fully establish a conclusive association between amino acids and diabetes. To address this limitation, we have added a sentence in Lines 302–303 in the Discussion and a phrase in Lines 334–336 in the Conclusion to emphasize the importance of conducting prospective longitudinal studies in the future.
Discussion: “In addition, a follow-up study in this population is expected to elucidate the effects on other glycemic biomarkers and the incidence of prediabetes and diabetes.”
Conclusion: “Future studies in other populations with different dietary patterns and a follow-up study in this population should be conducted to elucidate the effects of amino acid composition. ”

Reviewer 2 Report
Comments and Suggestions for Authors
Dear authors,
Thank you for giving me the opportunity to review your manuscript which, in an extended paper, studies the association between dietary amino acid composition and glycemic biomarkers in Japanese adolescents
The study is very interesting.
Here are my comments:
The study is very well documented and the large number of participants ensures the correctness of the statistical processing. ABSTRACT must be written according to the templates. Its structure must be highlighted The INTRODUCTION chapter is very well written. The hypothesis and purpose are highlighted very well.
In the METHODS chapter, information is presented very well. Please explain in more detail the criteria by which adolescents were accepted into the study. RESULTS chapter
In mathematics, the arithmetic mean of a list of non-negative real numbers is equal to the geometric mean of the same list if and only if all the numbers in the list are the same. How do you explain the equality of these means in table 2? Are all the values collected from the adolescents in the study equal?!?!?!? The DISCUSSION chapter is well written. I appreciate that the limitations of the article are well presented.
The CONCLUSIONS chapter does not analyze the impact of this study on the scientific and medical community.
Good luck!
Author Response
March 11th, 2024
Dear Reviewer 2,
We appreciate your comments with positive suggestions and advice. We have responded to each of the comments, added new figures, and amended the manuscript and supplementary files, in conjunction the other reviewer’s comments. Corrected sentences have been yellow-highlighted in the revised manuscript.
I, on behalf of the coauthors, wish to resubmit the manuscript. We hope that the manuscript is revised according to the reviewers’ intentions and is now suitable for publication.
Sincerely,
Masayuki Okuda
- The study is very well documented, and the large number of participants ensures the correctness of the statistical processing.
Response: Thank you for your encouraging comment.
- ABSTRACT must be written according to the templates. Its structure must be highlighted
Response: We have revised the Abstract according to the journal requirement: “the Abstract with about 200 words in a single paragraph with the style of structured abstracts, but without headings.”
“Abstract: Protein intake reportedly increases the risk of diabetes; however, the results have been inconsistent. Diabetes in adulthood may be attributed to early life dietary amino acid composition. This study aimed to investigate the association between amino acid composition and glycemic biomarkers in adolescents. Dietary intake was assessed using a food frequency questionnaire, and fasting glucose and insulin levels were measured in 1,238 eighth-graders. The homeostatic model assessment (HOMA) indices (insulin resistance and β-cell function) were calculated. Anthropometrics were measured and other covariates were obtained from a questionnaire. Amino acid composition was isometric log-transformed according to the compositional data analysis, which was used as explanatory variables in multivariate linear regression models for glucose, insulin, and HOMA indices. Only the association between glucose and leucine was significant. In replacement of other amino acids with leucine, an increase of 0.1% of total amino acids correlated with a lower glucose level (-1.02 mg/dl). One-to-one substitution of leucine for isoleucine or methionine decreased glucose (-2.98 and -2.28 mg/dl, respectively). Associations with other biomarkers were not observed. In conclusion, compositional data analysis of amino acids revealed an association only with glucose in adolescents; however, the results of this study should be verified in other populations.
- The INTRODUCTION chapter is very well written. The hypothesis and purpose are highlighted very well.
Response: Thank you for your positive comment.
- In the METHODS chapter, information is presented very well. Please explain in more detail the criteria by which adolescents were accepted into the study.
Response: We have revised Figure 1 to clarify the criteria, and the following sentence has been added to clarify the ethical considerations in Lines 75–79, and exclusion criteria in Lines 121–127. The number of schools has been added in Line 73.
“The students were encouraged to participate in the project organized by the Shunan City educational board and schools, and both they and their guardians provided written informed ascent and consent, respectively, to participate in the study, allowing the collection of their samples by their own volition.” Lines 75–79.
“From 4,101 students who attended junior high schools, 3,524 students gave informed consent. Serum samples from 44.5% (1,447/3,254) of adolescents who participated in the study were available for insulin measurement. After excluding participants who did not answer either questionnaire (n = 15), who had implausible energy intake (n = 40), with missing items of the questionnaire (n = 150), or with physician-diagnosed diseases (any of diabetes mellitus, dyslipidemia, hypertension, heart disease, and kidney disease; n = 4), data from 1,238 healthy participants were analyzed.” Lines 121–127.
“Eighth-grade students attending one of all 16 junior high schools in Shuna City, Japan” Line 73.
RESULTS chapter
- In mathematics, the arithmetic mean of a list of non-negative real numbers is equal to the geometric mean of the same list if and only if all the numbers in the list are the same. How do you explain the equality of these means in table 2? Are all the values collected from the adolescents in the study equal?!?!?!?
Response: We partially agree with your observation. Amino acids expressed as components were not equal among students; however, their variation was relatively small. The values of the coefficient of variance ranged from 0.018 to 0.091. The calculated means exhibited almost identical values with two decimal places. We have verified the functions used to calculate them: exp(mean(log(x)) and mean(x). To address this point, we have modified the sentence in Lines 179-180.
“the geometric means are similar to the arithmetic ones with coefficients of variance of 0.018–0.091.”
- The DISCUSSION chapter is well written. I appreciate that the limitations of the article are well presented.
Response: Thank you for your positive comment.
- The CONCLUSIONS chapter does not analyze the impact of this study on the scientific and medical community.
Response: According to the valuable suggestion, we have made appropriate revisions to the Conclusion section. In these revisions, we propose further research directions, particularly focusing on the distinct investigation of each amino acid and addressing the limitations of our study. in Lines 324–336.
“The effects on glucose varied among amino acids within the same group of BCAAs, which could not simply explain the collective effects of amino acid groups such as BCAAs. Therefore, the effects of each amino acid on health risks should be distinctively investigated. Given that the proportions of amino acids in the diet are influenced by each other, compositional data analysis could reveal causality between amino acids and health status by considering the interrelationship of amino acids. These findings underscore the importance of understanding amino acid composition in dietary management and preventive strategies for metabolic disorders in adolescents. In this study of Japanese adolescents, only the effect on glucose, but not the other biomarkers, was observed; thus, the causal relationship to diabetes remains unclear. Future studies in other populations with different dietary patterns and a follow-up study in this population should be conducted to elucidate the effects of amino acid composition. ”
